# First Polyphasic Study of Cheffia Reservoir (Algeria) Cyanobacteria Isolates Reveals Toxic Picocyanobacteria Genotype

**DOI:** 10.3390/microorganisms11112664

**Published:** 2023-10-30

**Authors:** Lamia Benredjem, João Morais, Guilherme Scotta Hentschke, Akila Abdi, Hajira Berredjem, Vitor Vasconcelos

**Affiliations:** 1Department of Molecular and Cellular Biology, Faculty of Natural and Life Sciences, Abbes Laghrour University of Khenchela, BP 1252 Road of Batna, Khenchela 40004, Algeria; benredjem.lamia@univ-khenchela.dz; 2Laboratory of Applied Biochemistry and Microbiology, Department of Biochemistry, Faculty of Sciences, Badji Mokhtar University, BP 12, Annaba 23000, Algeria; akila.abdi@yahoo.com (A.A.); h_berjem@yahoo.fr (H.B.); 3CIIMAR/CIMAR, Interdisciplinary Centre of Marine and Environmental Research, Terminal de Cruzeiros do Porto de Leixões, University of Porto, 4450-208 Matosinhos, Portugal; jmorais@ciimar.up.pt (J.M.); guilherme.scotta@gmail.com (G.S.H.); 4Departamento de Biologia, Faculdade de Ciências, Universidade do Porto, Rua do Campo Alegre, Edifício FC4, 4169-007 Porto, Portugal

**Keywords:** Algeria, cyanobacteria, *mcyE* gene, morphology, picocyanobacteria, phylogeny

## Abstract

Monitoring water supply requires, among other quality indicators, the identification of the cyanobacteria community and taking into account their potential impact in terms of water quality. In this work, cyanobacteria strains were isolated from the Cheffia Reservoir and identified based on morphological features, the 16S rRNA gene, phylogenetic analysis, and toxin production by polymerase chain reaction PCR screening of the genes involved in the biosynthesis of cyanotoxins (*mcyA*, *mcyE*, *sxtA*, *sxtG*, *sxtI*, *cyrJ*, and *anaC*). Thirteen strains representing six different genera: *Aphanothece*, *Microcystis*, *Geitlerinema*, *Lyngbya*, *Microcoleus*, and *Pseudanabaena* were obtained. The results demonstrated the importance of morphological features in determining the genus or the species when incongruence between the morphological and phylogenetic analysis occurs and only the utility of the 16S rRNA gene in determining higher taxonomic levels. The phylogenetic analysis confirmed the polyphyly of cyanobacteria for the *Microcystis* and Oscillatoriales genera. Unexpectedly, *Aphanothece* sp. CR 11 had the genetic potential to produce microcystins. Our study gives new insight into species with picoplanktonic (or small) cell size and potentially toxic genotypes in this ecosystem. Thus, conventional water treatment methods in this ecosystem have to be adapted, indicating the requirement for pre-treatment methods that can effectively eliminate picocyanobacteria while preserving cell integrity to prevent toxin release.

## 1. Introduction

Cyanobacteria are photosynthetic prokaryotes found in aquatic and terrestrial habitats [1]. These organisms are prolific producers of natural products recognized as toxins with hepatotoxic, neurotoxic, and dermatotoxic effects and are harmful to humans and animals [2]. For people, the Caruraru syndrome that occurred in Brazil (Caruaru City) was the most severe event. During this incident, 76 patients died in a hemodialysis clinic; analysis of the liver tissue and serum of the victims led to the identification of microcystins (MCs, hepatotoxic cyanotoxins) [3]. Nowadays, the monitoring of water intended for public use takes account of the identification of toxic cyanobacteria [4]. Moreover, the elimination of cyanobacteria in drinking water treatment services depends on the cyanobacterial species and the treatment procedures employed [5]. For instance, *Aphanezomenon* cells exhibit a higher propensity to traverse traditional sand filtration systems, leading to the discharge of intracellular toxins into treated drinking water [6].

Morphological analyses have been the basis of cyanobacterial classification systems [7]. However, because of limitations due to the extreme phenotypic flexibility depending on different environmental and culture conditions [8], a polyphasic approach combining morphological and molecular information has been used for accurate cyanobacteria identification [9]. Nevertheless, this approach cannot differentiate between toxic and nontoxic cyanobacteria strains. Accordingly, various immunological and analytical approaches, such as enzyme-linked immunosorbent assays (ELISA), the protein phosphatase inhibition (PPI) assay, chromatography, and mass spectrometry [4], have been used for this objective. However, sensitivity, specificity, and high-cost limitations have made them inadequate for routine use [10]. The capacity of cyanobacteria to produce toxins is associated with specific metabolic processes that are encoded by complex gene operons. Toxin-related genes PCR amplification is a reliable technique for detecting potential toxic strains [11].

In Algeria, several reports have described the occurrence of cyanobacteria in ecosystems with socio-economic importance [12,13,14]. The Cheffia Reservoir is a water body in northeastern Algeria; the water is used for drinking and irrigation. Previous reports have been limited to analyzing environmental samples and toxins in this ecosystem and identifying cyanobacteria by a culture-independent technique [12]. The cultivation approach has advantages over the cultivation-independent studies of cyanobacteria; it allows for detailed phenotypic, genetic, physiological, and biochemical characterization [15]. Genetic research with axenic cultures is critical for cyanobacteria polyphasic taxonomy investigations, particularly for understudied polyphyletic taxa [16]. Recent studies have shown that numerous newly described cyanobacteria are exclusively based on isolated strains [17,18,19,20]. Thus, the aim of this study was focused specifically on cyanobacterial isolates from the Cheffia Reservoir. Isolated strains were identified using morphological features and phylogenetic analysis. Furthermore, toxin-related genes were localized using specific primers.

## 2. Materials and Methods

### 2.1. Collection of Cyanobacterial Samples and Isolation

The cyanobacteria sample was collected from the Cheffia Reservoir (36°36′33.5″ N 8°02′34.8″ E, North-East of Algeria, Figure 1) using a plankton net (20 µm mesh size) on 21 October 2018. Cyanobacteria isolation was done by the enrichment of cyanobacterial environmental samples in a liquid culture medium, then plating by streaking each sample on a 1.5% agar culture medium. The plates were observed regularly under an inverted microscope Leica^®^ DMi8 (Leica Microsystems, Wetzlar, Germany), and isolated strains were transferred to new plates and also maintained on a liquid medium. The cyanobacteria were isolated, grown, and maintained in BG11 medium [21] under a 14:10 h light/dark cycle at 19 ± 1 °C. Cycloheximide (Sigma, Dorse, UK) at 100 μg mL^−1^ was added to the culture medium in the first isolation steps to inhibit the growth of eukaryotic microorganisms [22].

### 2.2. Morphological Characterization

Morphological studies of isolated strains were carried out under an Olympus BX51 light microscope at 100×, 400×, and 1000× magnifications. In addition, photographs were taken with a Leica^®^ DFC550 digital camera, and measurements were determined for each strain (n = 100 cells) at 1000× magnification using Leica^®^ LAS X V.4 software. The strains were classified according to the morphological criteria of Komárek and Anagnostidis [24,25].

### 2.3. DNA Extraction, Amplification (PCR) and Sequencing

Fresh cyanobacterial cultures at the exponential growth phase were harvested by centrifugation (Micro STAR 17R, VWR) at 12.000× *g* for 5 min and stored at −20 °C. Genomic DNA (gDNA) was extracted using the Purelink Genomic DNA Mini Kit (Invitrogen, San Diego, CA, USA) according to the manufacturer’s instructions for Gram-negative bacteria. The integrity and quality of DNA were checked on 1.0% agarose gel, and gDNA was stored at −20 °C. The 16S rRNA gene was amplified using the primers CYA359F (GGGGAATYTTCCGCAATGGG), CYA781R (GACTACTGGGGTATCTAATCCCATT) [26], 27F1 (5′-AGAGTTTGATCCTGGCTCAG3′) and 1494Rc (5′-TACGGCTAC CTTGTTACGAC-3′) [27]. The PCRs reactions were performed with a Biometra TProfessional gradient thermocycler (Biometra, Göttingen, Germany) in a final volume of 20 μL following the methodologies described previously by Nübel et al. [26] and Neilan et al. [27]. The reaction mixture contained 4 μL 5× Green GoTaq^®^ Flexi buffer (Promega, Madison, WI, USA), 2 μL MgCl_2_ (25 mM), 2 μL of each primer (10 μM), 1 μL of deoxynucleoside triphosphate (dNTP, 10 μM) mix (Promega, Madison, WI, USA), bovine serum albumin (BSA), 0.1 μL of GoTaq^®^ DNA polymerase (Promega) and 1 μL of template DNA. PCR products were examined by electrophoresis in 1.0% agarose gels stained with SYBR^®^ Safe DNA Gel Stain (Thermo Fisher Scientific, Carlsbad, CA, USA) and then purified with a NucleoSpin^®^ Gel and PCR Clean-up Kit (Macherey-Nagel, Düren, Germany) according to the manufacturer’s instructions. The obtained products were sequenced bidirectionally by Sanger sequencing by GATC Biotech (Ebersberg, Germany).

For the obtained sequences with low DNA quality, purified PCR products were cloned into the pCR™2.1-TOPO^®^ vector (Invitrogen TOPO^®^ TA Cloning Kit) and then transformed into *Escherichia coli* OneShot^®^ TOP10 cells (Invitrogen) following the instructions of the manufacturers. After the white-blue selection on the ampicillin 1.5% agarose plates with Luria Bertani (LB) medium, the colonies were transferred into fresh liquid LB medium with 100 μg ml^–1^ of ampicillin and cultured overnight at 37 °C with shaking at 2000 rpm. The Plasmid DNA was isolated using GenElute™ Plasmid Miniprep Kit (Sigma-Aldrich, St. Louis, MO, USA) according to the manufacturer’s instructions. The plasmids were sequenced bidirectionally, using the following primers: 27F1, CYA359F, CYA781R, and 1494Rc, by Sanger sequencing at GATC Biotech (Ebersberg, Germany).

### 2.4. Sequence Analysis

Raw forward and reverse sequences were assembled using Geneious (version 8.1.8, Biomatters Ltd., Auckland, New Zealand) and then deposited in the GenBank database “http://www.ncbi.nlm.nih.gov/ (accessed on 26 January 2022)” under the accession numbers OM401262 to OM401274. The obtained sequences were compared to previously published sequences in NCBI (National Center for Biotechnology Information; accessible via http://www.ncbi.nlm.nih.gov/) using the BLASTN algorithm. Based on the resulting blast analysis, a phylogenetic tree was constructed using the “One Click” mode of the online web service Phylogeny.fr platform (http://www.phylogeny.fr/simple_phylogeny.cgi [28]).

### 2.5. PCR Amplification for Toxin Production

PCR was used to amplify toxin-synthesis-related genes encoding microcystin (*mcyA*), the amino transferase (AMT) of the microcystin and nodularin synthetases complexes (*mcyE*), saxitoxin (*sxtA*, *sxtG*, and *sxtI*), anatoxin (*anaC*), and cylindrospermopsin (*cyrJ*) by using specific primer sets (Table 1) and PCR programs described previously [29,30,31,32,33,34,35]. The PCR reactions were performed in a total volume of 18 µL using the GoTaq^®^ DNA polymerase (Promega, Madison, WI, USA) 1× PCR buffer, 2.5 mM MgCl_2_, 250 µM of dNTP mix, 10 pmol of each primer, 0.5 U of Taq DNA polymerase and 1 µL of genomic DNA. The PCRs were run either on a Veriti thermal cycler (Applied Biosystems, Foster City, CA, USA) or a Biometra TProfessional gradient thermocycler (Biometra, Göttingen, Germany). Obtained PCR products were examined by electrophoresis in 1.0% agarose gels stained with SYBR^®^ Safe DNA Gel Stain (Thermo Fisher Scientific, Carlsbad, CA, USA) and photographed under UV transillumination. *Microcystis aeruginosa* LEGE 91339 was used as a positive control for the *mcyA* and *mcyE* genes, *Aphanizomenon gracile* LMECYA 040 for the *sxtA*, *sxtG* and *sxtI* genes, *Cylindrospermopsis raciborskii* LEGE 97047 and *Anabaena* sp. LEGE X-002 for the *cyrJ* and *anaF* genes, respectively. These strains belong to the culture collection of the Blue Biotechnology and Ecotoxicology Laboratory (Blue Biotechnology and Ecotoxicology Culture Collection, LEGE-CC) of CIIMAR (Porto, Portugal).

## 3. Results

### 3.1. Cyanobacterial Isolation and Morphological Identification

In this work, thirteen cyanobacteria isolates were obtained. Following the morphological features of Komárek and Anagnostidis [23,24], the isolated strains were identified as *Geitlerinema* (*G.*) *amphibium* (four strains), *Lyngbya* (*L.*) *nigra* (one strain), *L. stagnina* (one strain), *L. cincinnata* (one strain), *Microcoleus* sp. (one strain), *Pseudanabaena* (*P.*) *rosea* (one strain), *Aphanothece* sp. (2 strains), *Microcystis (M.*) *flos-aquae* (one strain) and *Microcystis* sp. (one strain). Overall, 69% of the isolates were identified morphologically at the species level, and 31% were identified only at the genus level. The morphological features of isolates are illustrated in Table 2 and Table 3, and Figure 2 and Figure 3 represent their photomicrographs.

### 3.2. Phylogenetic Study

BLAST analysis showed high sequence similarities ranging from 97.26 to 99.91% between the 16S rRNA gene sequences of cyanobacterial isolates from the Cheffia Reservoir and 27 strain sequences from the Chroococcales, Oscillatoriales, and Synechococcales orders accessible in GenBank (Table 4).

The phylogenetic tree obtained with the studied strains is shown in Figure 4. The tree included five main clusters. Cluster A comprised sequences of *Microcoleus* sp. and *Lyngbya* spp. Sub-cluster I included *Microcoleus* sp. CR8 isolate and *Microcoleus* HTT-U-KK5 strain deposited in GenBank. Sub-cluster II contained *Lyngbya* isolates with sequences from the same genera: *L. martensiana* H3b/33 and two other sequences from *Oscillatoria* UIC 10045 and *Phormidium* cf. *irriguum* CCALA 759. Cluster B consists only of the *Microcystis flos-aquae* CR13 isolated in this work with sequences from the same species, *M. flos-aquae* CHAB541, and those of *Synechocystis* SAG 45.90 and *Sphaerovacum brasiliense* CCIBt3094, from freshwater eutrophic reservoirs and seawater from Europe, Asia, and South America, deposited in GenBank. Cluster C is represented by *G. amphibium* strains with the strain *G. amphibium* and *Limnothrix*, *Jaaginema*, *Pseudanabaena*, and *Anagnostidinema* sequences. Cluster D included the two strains of the picocyanobacterial genera *Aphanothece* and the isolate identified morphologically as *Microcystis* sp. grouped with sequences belonging to *Cyanobacterium*, *Cyanobium*, *Synechococcus*, *Aphanothece*, *Cyanodictyon*, *Aphanocapsa*, and *Microcystis* deposited in GenBank. Finally, cluster E included the *Psedanabaena* isolate and sequences in the GenBank database of the terrestrial and freshwater species: *P. limnetica* Lim1, *L. redekei* CCAP 1459/29 and *Arthronema gygaxiana* UTCC 393 isolated from France, Japan, and Italy.

### 3.3. PCR Amplification of Toxin-Encoding Genes

The molecular screening of the genes involved in the biosynthesis of microcystin was positive for the *mcyE*-specific microcystin/nodularin gene for *Aphanothece* sp. CR11 strain, as shown in Figure 5. However, none of the isolated strains tested positive for anatoxin (*anaC*), saxitoxin (*sxtA*, *sxtG*, and *sxtI*), or cylindrospermopsin (*cyrJ*) genes.

## 4. Discussion

In order to characterize cyanobacteria isolates from the Cheffia Reservoir, morphological, molecular and phylogenetic analyses were conducted. The morphological identification up to the species level was successful for some strains. More importantly, the strains from the genera *Geitlerinema*, *Lyngbya* and *Microcystis* showed substantial phenotypic plasticity. All the isolated *Geitlerinema* strains have identical morphological characteristics and are similar to *G. amphibium* described by Bittencourt-Oliveira et al. [36]. However, our isolates vary from these organisms; their cell length was less than these organisms, which are 2.2–7 µm long, and their cell width was relatively wider than these organisms, which are 1.0–2.2 µm wide. The cells of isolated strains belonging to the species *P. rosea*, *L. nigra*, *L. cincinnata*, and *L. stagnina* exhibited a larger width than those described by Komárek and Anagnostidis [24], which are 1.61–2.67, 12.82–17.84, 13.46–19.66 and 13.00–19.36 µm, respectively. As well, the *M. flos-aquae* strain had cells with a larger diameter than the *M. flos-aquae* described by Komárek and Anagnostidis [25], which is 3.5–4.8 µm. The genus *Microcystis* has previously shown significant morphological flexibility [37,38,39]. This morphological plasticity has been related to several environmental or cultivation factors, such as medium composition, temperature, and light intensity [40,41].

Our findings also showed that morphological and phylogenetic classifications might be incompatible. This is the case of the strain *Microcystis* sp. CR 12, assigned to the genus *Microcystis* based on the morphological description of this genus in Bergey’s Manual of Determinative Bacteriology [42]. The genus *Microcystis* is characterized by the following features: coccoid cells with aerotopes and a tendency to form colonies delimited by mucilage. Differences between molecular and morphological descriptions have been extensively reported [43,44,45,46]. Moreover, it has also been demonstrated that GenBank contains erroneously classified species. Strains in culture collections have also been misnamed, and they may be found in GenBank and other culture collections under various names [43]. Moreover, with the numerous revisions to cyanobacteria taxonomy, there is no simple way to update existing databases [47], generating even more difficulty for cyanobacteria assignment [30,44]. Nonetheless, the number of 16S rRNA gene sequences now accessible in global databases for some species is still low, and additional systematic efforts will be necessary to clarify existing taxonomic ambiguities [30].

The isolated *Lyngbya* strains were morphologically assigned to three distinct species, although phylogenetically related to one species, *L. martensiana* H3b/33. The reliability of 16S rRNA gene sequences to identify *Lyngbya* strains at the species level and lower has been contested [48]. Engene et al. [49] suggested only using 16S rRNA gene sequences to identify *Lyngbya* strains at the genus level. Furthermore, Fathalli et al. [50] confirmed that even when closely related species are morphologically different, it may not be possible to differentiate them properly by the 16S rRNA gene. The 16S rRNA gene is less effective for phylogenetic investigations of closely related organisms due to its conserved character and lower evolutionary rate fluctuation than protein-encoding genes [51].

Three distinct clades were obtained from the Oscillatoriales strains, confirming the polyphyletic origin of this order [52]. The obtained *Microcystis* morphospecies were mixed in the phylogenetic tree and cannot be identified as monophyletic organisms, which also confirms the polyphyly of the Chroococcales [16,53].

Our study represents the first report of a potentially toxic genotype from the genus *Aphanothece* in this ecosystem. Picocyanobacteria MCs producers have been isolated from the Tabocas reservoir in the city of Caruaru, where the first human poisoning episode was reported [54]. MCs are the most dangerous group of cyanotoxins; they inhibit serine and threonine phosphatases—PP1 and PP2A—causing significant hepatotoxicity and acting as a carcinogen [55].

Some studies have also shown that strains of picocyanobacteria can produce MCs in culture, but no study has explored the presence of toxicity genes [56,57]. The picocyanobacteria group can also produce blooms responsible for the significant loss of benthic wildlife [57,58,59,60]. The most studied picocyanobacteria blooms have been observed in the northern Mediterranean at the Comacchio lagoons (northwest Adriatic coast) [61]. These blooms led to the death of bottom flora and benthic fauna and the loss of valuable resources of fish (eel and mullet) and mollusks [62]. Śliwińska-Wilczewska et al. [57] and Felpeto et al. [63] suggest that picocyanobacteria blooms are a new phenomenon that requires comprehensive studies. Although the impacts of climate change on picocyanobacterial blooms are diverse, most current evidence suggests that this process increases the amplitude and frequency of these events [57,64]. So far, little is known regarding the toxicity of picocyanobacteria, whereas the number of reports about their prevalence in ecosystems is growing. Therefore, the problem of picocyanobacteria toxicity requires more attention and interest from researchers [57].

## 5. Conclusions

The results highlighted that identifying cyanobacteria isolates by combining morphological and molecular methods is still challenging. Furthermore, for the first time in this reservoir, a potentially toxic genotype from picocyanobacteria was described; this group of cyanobacteria is often overlooked, although its toxicity is of great importance. Thus, water treatment methods in this ecosystem must consider the presence of picocyanobacteria strain. This adaptation entails the implementation of pre-treatment techniques that can selectively eradicate picocyanobacteria while preserving the integrity of their cellular structure and therefore, the release of ovoid cyanotoxins. In addition, it is vitally important to conduct in-depth research on the function of picocyanobacteria in aquatic ecosystems.

## Figures and Tables

**Figure 1 microorganisms-11-02664-f001:**
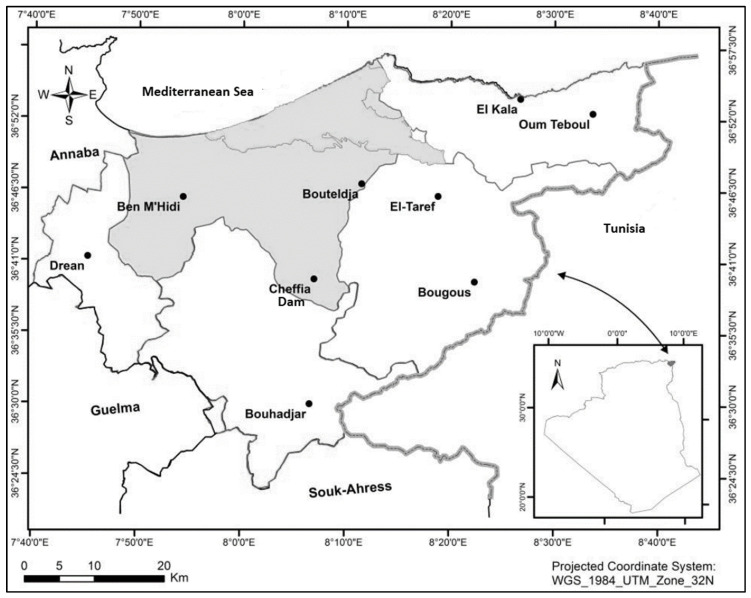
Localization of Cheffia Reservoir (Adapted) [23].

**Figure 2 microorganisms-11-02664-f002:**
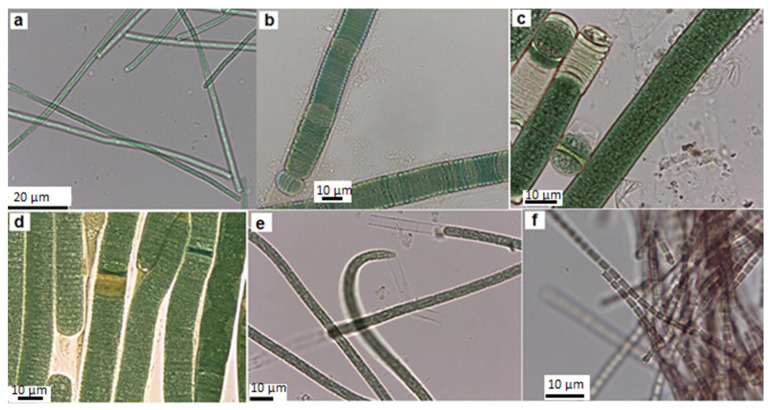
Light microscope images of the filamentous isolated strain from the Cheffia Reservoir. (**a**) *G. amphibium* CR1, (**b**) *L. nigra* CR5, (**c**) *L. stagnina* CR6, (**d**) *L. cincinnata* CR7, (**e**) *Microcoleus* sp. CR8, and (**f**) *P. rosea* CR9.

**Figure 3 microorganisms-11-02664-f003:**
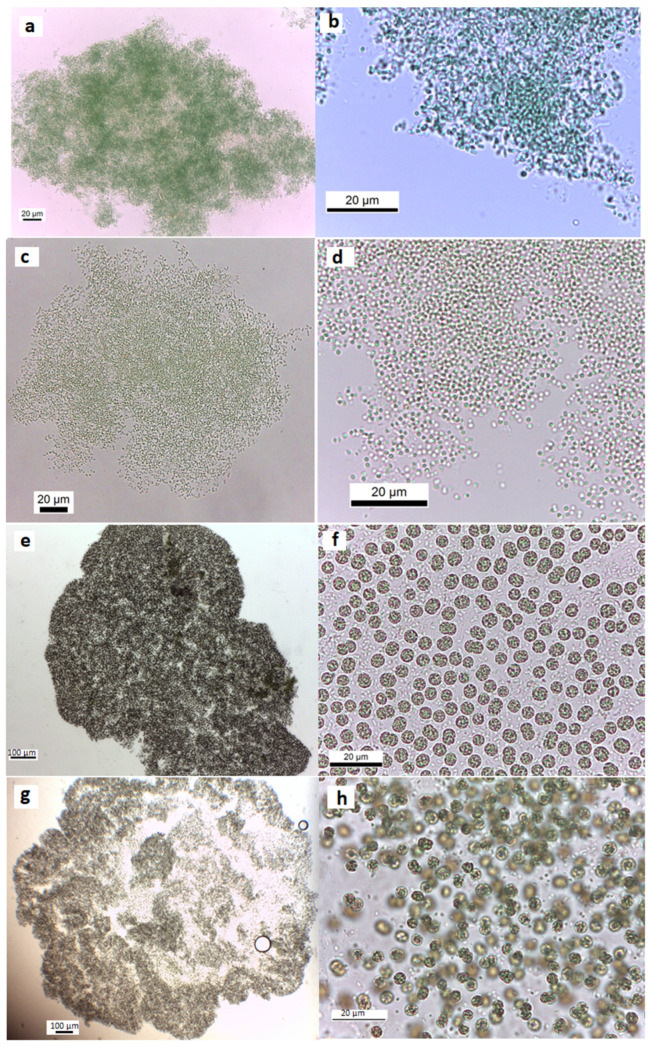
Light microscope images of the colonial isolated strains from the Cheffia Reservoir. (**a**,**b**) *Aphanothece* sp. CR10, (**c**,**d**) *Aphanothece* sp. CR11, (**e**,**f**) *Microcystis* sp. CR12, and (**g**,**h**) *M. flos-aquae* CR13.

**Figure 4 microorganisms-11-02664-f004:**
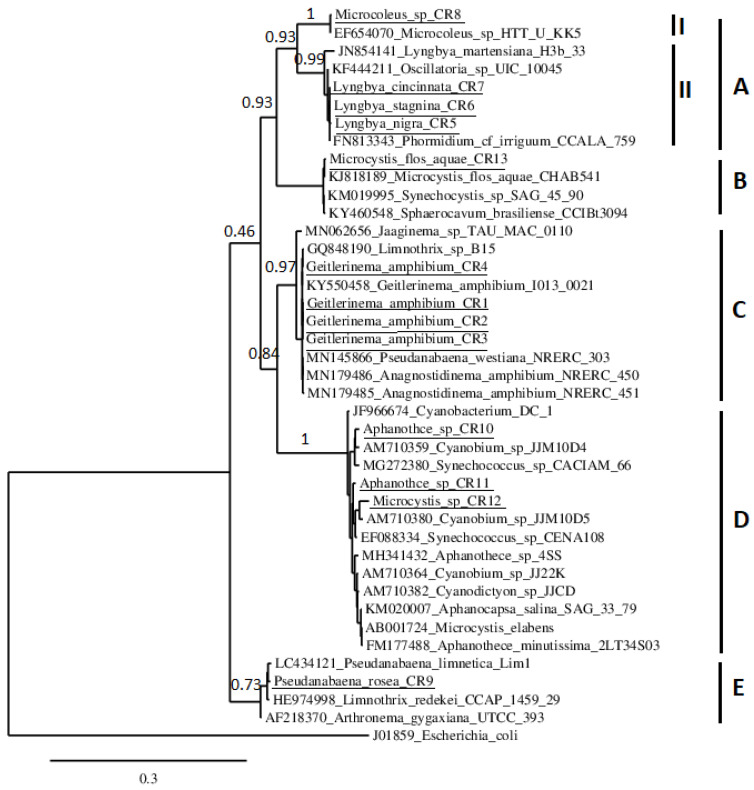
Phylogenetic relationships of Cheffia Reservoir cyanobacteria isolates with Genbank cyanobacteria strains based on 16S rRNA sequences. *Escherichia coli* (J01859) was used as the outgroup. Isolates from this study are underlined. A–E indicate the clusters; I, II indicate the sub-clusters.

**Figure 5 microorganisms-11-02664-f005:**
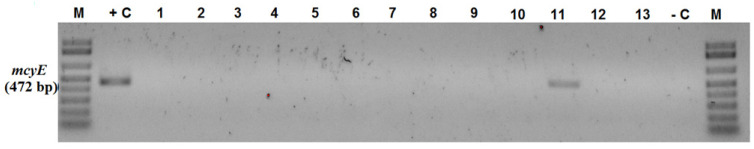
Gel electrophoresis of the *mcyE* gene fragments obtained from isolated strains from the Cheffia Reservoir. M: Marker (DNA ladder 1 kb), +C: positive control, *M. aeruginosa* LEGE 91339; Lane 1: *G. amphibium* CR1; Lane 2: *G. amphibium* CR2; Lane 3: *G. amphibium* CR3; Lane 4: *G. amphibium* CR4; Lane 5: *L. nigra* CR5; Lane 6: *L. stagnina* CR6; Lane 7: *L. cincinnata* CR7; Lane 8: *Microcoleus* sp. CR8; Lane 9: *P. rosea* CR9; Lane 10: *Aphanothece* sp. CR10; Lane 11: *Aphanothece* sp. CR11; Lane 12: *Microcystis* sp. CR12; Lane 13: *M. flos-aquae* CR13; -C: negative control.

**Table 1 microorganisms-11-02664-t001:** Primer sets used for PCR.

Target	Primer Pair	Sequence (5′−3′)	Size (bp)	Reference
*mcyA*	mcyA-Cd1F mcyA-Cd1R	AAAATTAAAAGCCGTATCAAA AAAAGTGTTTTATTAGCGGCTCAT	297	[35]
*mcyE*	HEPF HEPR	TTTGGGGTTAACTTTTTTGGGCATAGTC AATTCTTGAGGCTGTAAATCGGGTTT	472	[29]
*sxtA*	sxtA855F sxtA1480R	GACTCGGCTTGTTGCTTCCCC GCCAAACTCGCAACAGGAGAAGG	648	[34]
*sxtG*	sxtG432F sxtG928R	AATGGCAGATCGCAACCGCTAT ACATTCAACCCTGCCCATTCACT	519	[34]
*sxtI*	sxtI 682F sxtI 877R	GGATCTCAAAGAAGATGGCA GCCAAACGCAGTACCACTT	200	[30]
*cyrJ*	cynsulF cylnamR	ACTTCTCTCCTTTCCCTATC GAGTGAAAATGCGTAGAACTTG	584	[31]
*anaC*	anaC-genF anaC-genR	TCTGGTATTCAGTCCCCTCTAT CCCAATAGCCTGTCATCAA	366	[32]

**Table 2 microorganisms-11-02664-t002:** Morphological characteristics of the filamentous cyanobacterial strains isolated from the Cheffia Reservoir. L—length (µm); W—width (µm).

Isolates	Trichome Description	Cell Description	Dimensions (µm)
*G. amphibium* CR1; CR2; CR3; CR4	Bright blue-green trichomes, motile, without sheaths, more or less straight, not constricted at cross walls	Cells cylindrical	L = 1.44–4.73 W = 1.38–2.92
*L. nigra* CR5	Straight or slightly curved trichomes, constricted at the cross-walls, with calyptras, isopolar with sheaths. Sheaths thin and firm, open at the end, colorless or bluish	Cells are short, discoid, with granular content and aerotopes	L = 1.95–5.15 W = 12.20–15.93
*L. stagnina* CR6	Trichomes isopolar. Sheaths thin and firm, distinct, colorless, open at the ends. Filaments constricted at cross-walls and motile by gliding	Cells grayish blue-green, distinctly shorter than wide	L = 1.9–5.2 W = 13.00–19.36
*L. cincinnata* CR7	Cylindrical, slightly waved trichomes, constrictions at cross wall, isopolar. Sheaths firm, thick, colorless, lamellated and opened	Cells discoid, cell content is granulated	L = 2.09–4.62 W = 13.46–19.66
*Microcoleus* sp. CR8	Wavy or screw-like coiled trichomes, constricted at cross walls, motile, isopolar and delimited by firm and thickened sheaths, homogenous, open at the ends, hyaline	Cells mostly isodiametric or slightly longer than wide. Apical cells are conical without calyptras	L = 3.09–6.02 W = 3.08–5.17
*P. rosea* CR9	Straight or slightly curved trichomes, reddish violet, without sheath, with constrictions at the cross walls	Cells cylindrical without aerotopes	L = 1.67–3.89 W = 1.61–2.67

**Table 3 microorganisms-11-02664-t003:** Morphological characteristics of the colonial cyanobacterial strains isolated from the Cheffia Reservoir.

Isolates	Colony Description	Mucilage	Cell Description	Cell Diameter (µm)
*Aphanothece* sp. CR10	Irregular, microscopic to macroscopic colonies, blue-green	Thin, colorless mucilage and distinct at the margin	Cells mainly spherical or oval, with gas vesicle, bright blue-green content, and delimited with individual envelopes. Envelopes are firm, colored in dark blue-green	1.07–1.98
*Aphanothece* sp. CR11	Spherical colonies, rough in outline, microscopic, free-living, forming macroscopic granular agglomerations	Mucilage colorless and homogeneous, delimited at the margin, and follows the irregular outline of the colony, not diffluent, without a refractive outline	Cells spherical, pale or bright blue-green, slightly distant from one another, having fine granulation, without gas vesicles, and enveloped by thin individual layer	0.90–1.36
*Microcystis* sp. CR12	Colonies macroscopic, lenticular, slightly elongate, three-dimensional, agglomerated in macroscopic, free-floating, gelatinous, blue-green masses	Narrow, colorless mucilage, distinctly delimited along cell agglomerations and forming refractive outline	Cells spherical, densely aggregated, with individual envelopes, content is homogeneous, olive green or brownish with aerotopes	4.02–5.97
*M. flos-aquae* CR13	Spherical colonies, with irregular margins, microscopic to macroscopic, free-floating, compact, or clathrate, with densely irregularly arranged cells gathered in small agglomerations	Mucilage colorless, slightly distant from cell clusters, and delimited by slightly refractive outline	Cells spherical or hemispherical after division, with individual thick envelopes. Cell content appears granular, olive green, or brownish, with aerotopes	3.98–5.77

**Table 4 microorganisms-11-02664-t004:** 16S rRNA gene-sequence-based identity (%) between the cyanobacterial isolates from the Cheffia Reservoir and their closest match available in Genbank (NCBI).

Isolates	Closest Match (Accession Number)	Query Coverage (%)	Percent Identity (%)
*G. amphibium* CR1	*Geitlerinema amphibium* I013-0021 (KY550458)	82	99.39
	*Anagnostidinema amphibium* NRERC-450 (MN179486)	98	99.13
	*Pseudanabaena westiana* NRERC-303 (MN145866)	98	99.13
	*Limnothrix* sp. B15 (GQ848190)	98	99.13
*G. amphibium* CR2	*Geitlerinema amphibium* I013-0021 (KY550458)	83	99.74
	*Pseudanabaena westiana* NRERC-303 (MN145866)	99	99.56
	*Limnothrix* sp. B15 (GQ848190)	100	99.56
	*Anagnostidinema amphibium* NRERC-451 (MN179485)	100	99.49
	*Jaaginema* sp. TAU-MAC 0110 (MN062656)	96	97.59
*G. amphibium* CR3	*Geitlerinema amphibium* I013-0021 (KY550458)	85	99.91
	*Anagnostidinema amphibium* NRERC-450 (MN179486)	100	99.70
	*Pseudanabaena westiana* NRERC-303 (MN145866)	100	99.70
	*Limnothrix* sp. B15 (GQ848190)	100	99.70
	*Jaaginema* sp. TAU-MAC 0110 (MN062656)	97	97.71
*G. amphibium* CR4	*Geitlerinema amphibium* I013-0021 (KY550458)	85	99.91
	*Limnothrix* sp. B15 (GQ848190)	100	99.85
	*Anagnostidinema amphibium* NRERC-450 (MN179486)	100	99.70
	*Pseudanabaena westiana* NRERC-303 (MN145866)	100	99.70
	*Jaaginema* sp. TAU-MAC 0110 (MN062656)	98	97.73
*L. nigra* CR5	*Phormidium* cf. *irriguum* CCALA 759 (FN813343)	99	99.85
	*Oscillatoria* sp. UIC 10045 (KF444211)	93	99.35
	*Lyngbya martensiana* H3b/33 (JN854141)	95	97.78
*L. stagnina* CR6	*Phormidium* cf. *irriguum* CCALA 759 (FN813343)	97	99.85
	*Oscillatoria* sp. UIC 10045 (KF444211)	90	99.28
	*Lyngbya martensiana* H3b/33 (JN854141)	92	97.73
*L. cincinnata* CR7	*Phormidium* cf. *irriguum* CCALA 759 (FN813343)	97	99.63
	*Oscillatoria* sp. UIC 10045 (KF444211)	91	99.04
	*Lyngbya martensiana* H3b/33 (JN854141)	93	97.50
*Microcoleus* sp. CR8	Microcoleus sp. HTT-U-KK5 (EF654070)	99	99.40
*P. rosea* CR9	*Pseudanabaena limnetica* Lim1 (LC434121)	100	99.27
	*Limnothrix redekei* CCAP 1459/29 (HE974998)	100	99.05
	*Arthronema gygaxiana* UTCC 393 (AF218370)	100	98.39
*Aphanothece* sp. CR10	*Synechococcus* sp. CACIAM 66 (MG272380)	100	99.11
	*Cyanobium* sp. JJM10D4 (AM710359)	100	98.96
	*Aphanothece minutissima* 2LT34S03 (FM177488)	100	97.41
	*Microcystis elabens* (AB001724)	100	97.26
	*Aphanothece* sp. 4SS (MH341432)	86	97.79
*Aphanothece* sp. CR11	*Synechococcus* sp. CENA108 (EF088334)	100	98.96
	*Cyanobium* sp. JJ22K (AM710364)	100	98.30
	*Cyanobacterium* DC-1 (JF966674)	100	98.30
	*Cyanodictyon* sp. JJCD (AM710382)	100	98.22
	*Aphanocapsa salina* SAG 33.79 (KM020007)	100	98.07
	*Aphanothece* sp. 4SS (MH341432)	87	97.89
*Microcystis* sp. CR12	*Synechococcus* sp. CENA108 (EF088334)	100	98.08
	*Cyanobium* sp. JJM10D5 (AM710380)	100	98.00
*M. flos-aquae* CR13	*Microcystis flos-aquae* CHAB541 (KJ818189)	100	99.77
	*Sphaerocavum brasiliense* CCIBt3094 (KY460548)	100	99.32
	*Synechocystis* sp. SAG 45.90 (KM019995)	100	99.47

## Data Availability

Not applicable.

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
