# Peer review of "First Polyphasic Study of Cheffia Reservoir (Algeria) Cyanobacteria Isolates Reveals Toxic Picocyanobacteria Genotype"

_microorganisms, 2023, doi:10.3390/microorganisms11112664_

Round 1
Reviewer 1 Report
Comments and Suggestions for Authors
This work is devoted to the study of cyanobacteria in a reservoir in Algeria. Studying the biodiversity of cyanobacteria through cultivation is a labor-intensive process that requires skill and time, but it is this approach that allows us to discover new species or obtain new data on the physiology of species. This work could have greater scientific weight if some sections are corrected and supplemented. Comments on the tables are indicated in the article file, other comments are given in the Word file.

Author Response
Comments from Reviewer 1
This work is devoted to the study of cyanobacteria in a reservoir in Algeria. Studying the biodiversity of cyanobacteria through cultivation is a labor-intensive process that requires skill and time, but it is this approach that allows us to discover new species or obtain new data on the physiology of species. This work could have greater scientific weight if some sections are corrected and supplemented. Comments on the tables are indicated in the article file, other comments are given in the Word file.
Comments for Table 2
Comment 1: L. nigra CR5
1)The indicated width values of trichome are far exceeded the limits of the species diagnosis (8-11(14) μm), in addition, gas vacuoles are not characteristic for this species. Are you confident in your species identification?
2)Perhaps you measured the width of filament rather than trichome? - Please specify it in the table so as not to cause misunderstanding.
3)Is it the same in nature?
Response: we thank the reviewer for this comment and we agree that the mentioned value was the width of the filament. The width of the trichome is between 12.20 and 15.93 µm. It was corrected in the Table 2.
For the gas vacuoles, Gupta and Arora (1987) observed the formation of small vacuoles in L. nigra at a concentration of 8 µM of copper sulphate. We suggest that the morphological characteristics described in our study are related to the culture conditions. But, unfortunately, the study of morphological characteristics has not been carried out on strains in natural environment.
Reference: Gupta, A.B.; Arora, A. Morphology and physiology of Lyngbia nigra with reference to copper toxicity. Physiol. Plant. 1978, 44, 215-220. 10.1111/j.1399-3054.1978.tb08620.x
Comment 2: L. stagnina CR6
The indicated width values of trichome are far exceeded the limits of the species diagnosis (9.5-12 (13) μm). Are you confident in your species identification?
Response: we agree with the reviewer, the three Lyngbya strains exhibited a larger width than those described by Komárek & Anagnostidis, which could be related to the growth under laboratory conditions.
Comment 3: L. cincinnata CR7
it's a benthic species, it doesn't have aerotopes.
Response: the description of aerotopes was removed.
Comment 4: Microcoleus sp. CR8
it's a benthic species also, it doesn't have aerotopes. The arrangement of thylakoids in Microcoleus may look similar to aerotopes.
Response: description of aerotopes was removed.
Comment 5: P. rosea CR9
This species has very short trichomes up to 6 cells, which is inconsistent with the photo. I agree that it is Pseudanabaena, but I cannot agree that it is P. rosea.
Response: we agree with the reviewer that P. rosea has short trichomes, but all the other morphological characteristics are related to P. rosea. The number of cells observed in our study could be related to the culture conditions. Schipper et al. (2021) observed an increase in trichomes of Leptolyngbya sp. QUCCCM 56, both in number and in length, after 6 days of outdoor cultivation.
Reference: Schipper, K., Das, P., Al Muraikhi, M., AbdulQuadir, M., Thaher, M. I., Al Jabri, H. M. S. J., Wijffels, R. H., & Barbosa, M. J. (2021). Outdoor scale‐up of Leptolyngbya sp.: Effect of light intensity and inoculum volume on photoinhibition and oxidation. Biotechnology and Bioengineering, 1–12. https://doi.org/10.1002/bit.27750
Comment 6: Aphanothece sp. CR11
The morphology described by you with spherical cells corresponds more to the genus Aphanocapsa than to Aphanothece (in this genus the cells are elongated).
Response: we do not agree with the reviewer because for Aphanocapsa the cells are irregularly arranged according to Komárekand Johansen (2015) but for Aphanothece the cells are more or less evenly distributed within the mucilage.
Reference: Komárek, J. & J. R. Johansen, 2015. Coccoid Cyanobacteria. In Wehr, J. D., R. G. Sheath & R. P. Kociolek (eds), Freshwater Algae of North America: Ecology and Classification 2nd ed. Academic Press, Amsterdam: 75–134.
Comment 7: Hereafter, generic and species epithets are written in italics.
Response: all the genera and species are italicized, lines 24, 28, 45, 133, 162-164, 181-184, 203-204, 218-219, 237, 254, 282, 298-303, 309-311, 314-315, 317-318, 323, 325, 336-338, 349, and 358.
General comments
Response: All the comments in the manuscript are corrected, lines 32, 91, 219, 237, 238, 243, 247, 252, 302, and 392.

Reviewer 2 Report
Comments and Suggestions for Authors
Article title: First polyphasic study of Cheffia reservoir (Algeria) cyanobacteria isolates reveals toxic picocyanobacteria genotype
General comments:
Both Abstract and Introduction seem unfinished to me. Abstract gives only one short sentence about general values of the obtained results. Introduction then comes short when explaining novelty of the study and its importance for the topic. Please reconsider reformulation of both sections ends. Clearly state novelty and benefits of your work. You mentioned water treatment in abstract, then please suggest what modifications should be done to prevent microcystin intake…
Line 77: Use proper unit for Cycloheximide.
Line 98: change to MgCl2
Line 111: 100 “μg ml–1” Such unit does not exist. Please check all the units throughout the text and change to µg/mL or µg/mL-1
Line 137: Use italics when writing species full name Microcystis aeruginosa goes in italics. Do the same throughout the text, please.
Table 3: In table you declare range of cell diameters for each detected strain. Please add to the text hoe these values were determined. Did you use flow cytometry or cell counter or some graphical interface for images analysis or the ladder in the microscope etc?
Lines 232-233:” This morphological plasticity has been related to several environmental or cultivation factors.” Name some please with proper references.
Lines 285-286: “Thus, water treatment methods in this ecosystem must consider the presence of picocyanobacteria strains.” Please propose solution to such water treatment.
Conclusion section is very short and unconclusive which is in contrary to its purpose in the manuscript. Please consider rewriting.
Author Response
Comments from Reviewer 2
General comments:
Comment 1: Both Abstract and Introduction seem unfinished to me. Abstract gives only one short sentence about general values of the obtained results. Introduction then comes short when explaining novelty of the study and its importance for the topic. Please reconsider reformulation of both sections ends. Clearly state novelty and benefits of your work. You mentioned water treatment in abstract, then please suggest what modifications should be done to prevent microcystin intake…
Response: we agree with the reviewer’s suggestion and accordingly we have reformulated the end of abstract and introduction.
- For the abstract, we suggested the need for pre-treatment methods to remove intact picocyanobacterial cells, lines 31-32.
- For the introduction, we mentioned the importance of the identification of cyanobacteria for the selection of the appropriate water treatment strategy, lines 42-46.
Two references have been added in the section References as follows:
- lines 422-424: Zamyadi, A.; Glover, C.M.; Yasir, A.; Stuetz, R.; Newcombe, G.; Crosbie, N.D.; Lin, T.F.; & Henderson, R. Toxic cyanobacteria in water supply systems: Data analysis to map global challenges and demonstrate the benefits of multi-barrier treatment approaches. H2Open J. 2021, 4, 47-62. doi:10.2166/H2OJ.2021.067.
- lines 425-426: Zamyadi, A.; Dorner, S.; Sauve, S.; Ellis, D.; Bolduc, A.; Bastien, C.; Prevost, M. Species-dependence of cyanobacteria removal efficiency by different drinking water treatment processes. Water Res. 2013, 47, 2689–2700. doi:10.1016/j.watres.2013.02.040.
Comment 2: Line 77: Use proper unit for Cycloheximide.
Response: corrected, lines 91.
Comment 3: Line 98: change to MgCl2
Response: corrected, lines 123,156.
Comment 4: Line 111: 100 “μg ml–1” Such unit does not exist. Please check all the units throughout the text and change to µg/mL or µg mL-1
Response: we thank the reviewer for this comment, all the units have been corrected in the manuscript, lines 91-136.
Comment 5: Line 137: Use italics when writing species full name Microcystis aeruginosa goes in italics. Do the same throughout the text, please.
Response: all the genus and species manes have been adjusted in Italic font in the manuscript, lines 24, 28, 45, 133, 162-164, 181-184, 203-204, 218-219, 237, 254, 282, 298-303, 309-311, 314-315, 317-318, 323, 325, 336-338, 349, and 358.
Comment 6: Table 3: In table you declare range of cell diameters for each detected strain. Please add to the text how these values were determined. Did you use flow cytometry or cell counter or some graphical interface for images analysis or the ladder in the microscope etc?
Response: it is mentioned in the text that measurements were determined using Leica®Lasx software, section 2.2. Morphological Characterization, lines 105-106.
Comment 7: Lines 232-233:” This morphological plasticity has been related to several environmental or cultivation factors.” Name some please with proper references.
Response: we thank the reviewer for this suggestion. Factors related to the morphological plasticity, medium composition, temperature, and light intensity, have been cited, lines 320-321.
Tow references have been added in the section references as follows:
- lines 899-901: Zapomělová, E.; Hrouzek, P.; Řeháková, K.; Sabacká, M.; Stibal, M.; Caisová, L.; Komárková, J.; Lukesová, A. Morphological variability in selected heterocystous cyanobacterial strains as a response to varied temperature, light intensity and medium composition. Folia Microbiol. 2008, 53, 333–341. doi:10.1007/s12223-008-0052-8.
- lines 902-903: Miller, S.R.; Longley, R.; Hutchins, P.R.; Bauersachs, T. Cellular innovation of the cyanobacterial heterocyst by the adaptive loss of plasticity. Curr Biol. 2020, 20, 344-350. doi: 10.1016/j.cub.2019.11.056.
Comment 8: Lines 285-286: “Thus, water treatment methods in this ecosystem must consider the presence of picocyanobacteria strains.” Please propose solution to such water treatment.
Response: thank you for this comment. The application of pre-treatment for the removal of intact cells was proposed, lines 383-385.
Comment 9: Conclusion section is very short and unconclusive which is in contrary to its purpose in the manuscript. Please consider rewriting.
Response: the end of the conclusion has been reformulated, lines 383-385.
Reviewer 3 Report
Comments and Suggestions for Authors
The paper by Benredjem et al. focuses on the identification of potentially toxic cyanobacteria in the Cheffia Reservoir in Algeria. (It would not hurt to provide a map of its location as Figure 1).
The authors isolated thirteen strains from six different genera of cyanobacteria into cultures. They described their morphology using microscopy, isolated DNA and carried out phylogenetic analysis using the 16S rRNA gene, that is, they were identified, as is customary in similar studies. To identify genes involved in the biosynthesis of cyanotoxins, the authors used PCR screening. As a result of the research, the authors identified a strain that has the genetic potential to produce microcystins. The work was carried out at a modern level and is of practical importance for monitoring water quality. It is required to edit the text of the manuscript: Cheffia Reservoir - capitalize both words. If we are talking about a gene and not a toxin, then its name should be italicized. The names of genera and species of bacteria should be in italics. At the end of Abstract, remove the extra dot. In the names of subchapters (2.1, 2.2 and others), all words must be written in capital letters (like 2.3, for example). Correct the register in the designation of chemical substances MgCl2, and in the indication of the concentration of solutions -1. In Table 4 "cf." no need to put it in italics.The References section must be carefully reviewed and formatted in accordance with the journal's rules.
Author Response
Comments from Reviewer 3
The paper by Benredjem et al. focuses on the identification of potentially toxic cyanobacteria in the Cheffia Reservoir in Algeria. (It would not hurt to provide a map of its location as Figure 1).
The authors isolated thirteen strains from six different genera of cyanobacteria into cultures. They described their morphology using microscopy, isolated DNA and carried out phylogenetic analysis using the 16S rRNA gene, that is, they were identified, as is customary in similar studies. To identify genes involved in the biosynthesis of cyanotoxins, the authors used PCR screening. As a result of the research, the authors identified a strain that has the genetic potential to produce microcystins. The work was carried out at a modern level and is of practical importance for monitoring water quality.
We thank the Reviewer for his/her appreciation of our work.
A map of the Cheffia Reservoir localization has been added, lines 100 and 101.
Comment 1: It is required to edit the text of the manuscript: Cheffia Reservoir - capitalize both words.
Response: corrected, lines 2, 21, 69, 79, 84, 188, 202, 214, 218, 223, 231, 281, 298, 306 and 360.
Comment 2: If we are talking about a gene and not a toxin, then its name should be italicized.
Response: all the names of the genes have been italicized, lines 23, 33, 151, 153, 163-165, 286, and 288.
Comment 3: The names of genera and species of bacteria should be in italics.
Response: all the genera and species names have been italicized, lines lines 24, 28, 45, 133, 162-164, 181-184, 203-204, 218-219, 237, 254, 282, 298-303, 309-311, 314-315, 317-318, 323, 325, 336-338, 349, and 358.
Comment 4: At the end of Abstract, remove the extra dot.
Response: it has been removed, line 32.
Comment 5: In the names of subchapters (2.1, 2.2 and others), all words must be written in capital letters (like 2.3, for example).
Response: all the names of subchapters have been edited, lines 83, 102, 141, 150, 221, 284.
Comment 6: Correct the register in the designation of chemical substances MgCl2, and in the indication of the concentration of solutions -1.
Response: it has been corrected, lines 91, 123, 136, and 156.
Comment 7: In Table 4 "cf." no need to put it in italics.
Response: it has been corrected.
Comment 8: The References section must be carefully reviewed and formatted in accordance with the journal's rules.
Response: all the references have been adjusted according to the journal guidance, lines 413-1081.
Round 2
Reviewer 1 Report
Comments and Suggestions for Authors
Thank you to the authors for their detailed answers to all the questions that arose. The revision of figures and text was thorough and improved the work.
There was a misunderstanding: in the caption to figure 5 - is "Lane" a band or a Line? I can't find the word "Lane" in the dictionary.
Reviewer 2 Report
Comments and Suggestions for Authors
I have no additional requests or comments.